# Military AI Needs Technically-Informed Regulation to Safeguard AI Research and its Applications

**Riley Simmons-Edler**[*]
Department of Neurobiology,
Harvard Medical School &
Kempner Institute, Harvard University,
Boston, MA, USA.
riley_simmons-edler@hms.harvard.edu

**Jean Dong**[*]
Kennedy School of Government,
Harvard University,
Cambridge, MA, USA; &
Griffith University, QLD, AU
jeandong@hks.harvard.edu

**Paul Lushenko**[†]
Department of Military Strategy,
Planning, and Operations,
U.S. Army War College,
Carlisle, PA, USA.
pal243@cornell.edu[‡]

**Kanaka Rajan**[†]
Department of Neurobiology,
Harvard Medical School &
Kempner Institute, Harvard University,
Boston, MA, USA.
kanaka_rajan@hms.harvard.edu[§]

**Ryan P. Badman**[†]
Department of Neurobiology,
Harvard Medical School &
Kempner Institute, Harvard University,
Boston, MA, USA.
ryan_badman@hms.harvard.edu

## Abstract

Military weapon systems and command-and-control infrastructure augmented by artificial intelligence (AI) have seen rapid development and deployment in recent years. However, the sociotechnical impacts of AI on combat systems, military decision-making, and the norms of warfare have been understudied. We focus on a specific subset of lethal autonomous weapon systems (LAWS) that use AI for targeting or battlefield decisions. We refer to this subset as AI-powered lethal autonomous weapon systems (AI-LAWS) and argue that they introduce novel risks—including unanticipated escalation, poor reliability in unfamiliar environments, and erosion of human oversight—all of which threaten both military effectiveness and the openness of AI research. These risks cannot be addressed by high-level policy alone; effective regulation must be grounded in the technical behavior of AI models. We argue that AI researchers must be involved throughout the regulatory lifecycle. Thus, we propose a clear, behavior-based definition of AI-LAWS—systems that introduce unique risks through their use of modern AI—as a foundation for technically grounded regulation, given that existing frameworks do not distinguish them from conventional LAWS. Using this definition, we propose several technically-informed policy directions and invite greater participation from the AI research community in military AI policy discussions.

---

[*]equal contributions

[†]equal contributions

[‡]The views expressed in this article are those of the authors and do not reflect the official position of the U.S. Government, Department of Defense, Department of the Army, or the Army War College.

[§]corresponding author: Kanaka Rajan

39th Conference on Neural Information Processing Systems (NeurIPS 2025) Position Paper Track.

*Lay Abstract:*

We argue that recently developed military weapon and command systems using frontier AI are a categorically new type of weapon system with unique risks and regulatory needs. These systems have been deployed internationally in growing numbers and variety in recent years, and existing regulation and oversight mechanisms are insufficient and nonspecific. We discuss the importance of technical AI researchers contributing to this under-discussed and fast moving topic, and we outline specific definitions and policy proposals to aid future debate.

# 1 Introduction

Artificial intelligence (AI), also called machine learning (ML), is embedded in a growing number of deployed weapon systems. These include uncrewed aerial vehicles, uncrewed surface vessels and submarines, and battlefield coordination platforms that assist with targeting and decision support [1–11]. The rapid deployment of these AI-augmented systems has outpaced existing regulatory frameworks. Governance models for conventional lethal autonomous weapon systems (LAWS)—typically rule-based and non-adaptive—were not designed to address challenges posed by modern AI [8–11]. These challenges include opacity (hard-to-interpret decision processes), adaptivity (behavioral shifts in response to new inputs), and post-deployment drift (performance degradation over time) [2, 12–14].

Yet in both policy and AI communities, distinctions between conventional LAWS and modern AI-powered systems remain poorly defined—and are absent from existing treaties, principles, and deployment norms. These systems may misclassify targets under unfamiliar conditions, escalate conflicts due to unpredictable outputs, or obscure human accountability through black-box decisions—all while using AI models originally developed in civilian contexts (see Table 1).

As these systems become more central to warfare, civilian AI advances gain national security relevance, and governments may restrict the movement of people, research, and AI services to gain military advantage [12, 15]. Given the risks that reckless development of AI weapons pose, **we take the position that modern AI-powered autonomous weapon systems—what we call AI-LAWS—require distinct and novel regulations to avoid harms to AI research and the AI industry.** Effective regulation must consider the underlying ML technologies, and AI experts from diverse fields must be directly involved in the regulatory process.

To motivate this, we first review the current state of military AI development and policy in section 2, highlighting the range of systems publicly announced and already deployed. Next, we introduce a functional definition of AI-LAWS based on AI involvement in targeting, advising, or decision-making (section 3), which we use to identify systems that pose novel risks and require distinct regulatory consideration. Lastly, we propose several technically-informed policy directions grounded in this definition to address the risks AI-LAWS pose to both AI research and global security (section 4).

# 2 Background

We first survey deployed and in-development systems that meet our criteria for AI-LAWS (section 3), with examples listed in Table 2. We then examine the unique risks these systems pose in subsection 2.2, and explain why AI expert involvement is essential in their development, use, and regulation. These risks are not theoretical—they are structural features of how AI-LAWS are developed and behave, already visible in deployed systems [16]. Performance may degrade in new environments, behaviors may escalate tensions, and lethal decisions may be made by models no one fully understands. We summarize these risks in Table 1. Finally, we review existing regulatory efforts and alternate positions in subsection 2.3, and highlight why they fail to address risks specific to AI-LAWS.

## 2.1 Existing AI-LAWS

**AI-LAWS are already deployed worldwide.** Systems using AI to make real-time targeting or coordination decisions have already been demoed and deployed across air, land, sea, and command domains. Table 2 highlights examples with documented AI capabilities, from drone "swarms" to battlefield planning platforms. Many of these systems rely on models and tools developed in civilian research settings. Yet, researchers often remain unaware that their work may shape combat outcomes—or be deployed without rigorous validation or constraint [15, 17]. Some AI-LAWS have already seen combat [15, 10]. The Russian Lancet loitering munition, for example—a drone that

Table 1: *Example risks introduced by AI-LAWS that meet our oversight criteria.* Oversight must be grounded in technical behavior—not system intent—and informed by expertise from the AI research community. This list of risks is not intended to be comprehensive.

| Risk Type | Description | Example Scenario |
|---|---|---|
| Undetected failures in real-world use | Gaps in validation and overtrust in AI outputs make it harder to detect failures during deployment—e.g. in stressful or unfamiliar contexts. | Forest-trained targeting AI model misidentifies vehicles in deserts [27, 28], and commanders defer to faulty AI planning advice [29, 30]. |
| Opacity and black-box decision-making | AI-based targeting systems are difficult for human operators to understand or audit, causing unpredictable behavior in combat. | AI-enabled engagement system selects lethal target based on faulty sensors the operator cannot override in time, leading to unintended casualties [13]. |
| Erosion of AI research freedom and scientific exchange | Classified funding streams and dual-use constraints limit openness, autonomy, and international collaboration. | University AI lab begins operating under military classification rules, limiting publication and/or international collaboration [31, 12, 32]. |
| Channeling AI expertise towards military applications | Civilian AI researchers are recruited into military programs, sometimes without clear disclosure or opt-out mechanisms. | University scientists find their programs absorbed into a defense-funded lab, shifting away from open-source AI research [33, 31]. |
| Acceleration of arms races and regional or global instability | Widespread access to AI-LAWS lowers barriers to conflict escalation. | Deploying drone swarms without adequate testing/safeguards, risking conflict escalation [34, 35]. |

independently identifies and crashes into targets—reportedly includes AI-based autonomous targeting capabilities and has seen wide deployment in Ukraine [18–20]. AI-LAWS span a wide range of platforms, from disposable drones to naval vessels and advanced aircraft [21] (Table 2). These systems are being developed by both major and minor powers—with many systems being actively deployed [22, 23, 15]. For example, the Estonian Milrem THeMIS UGV, which includes autonomous navigation and potentially targeting, has seen battlefield use in Ukraine [24, 25]. Given the diversity of actors, platforms, and missions, even well-intentioned policies will fall short unless they account for the varied ways AI models are used across military operations [10, 26].

## 2.2 AI-LAWS present unique risks

**AI-LAWS create distinct risks across military, geopolitical, and institutional domains.** AI-LAWS pose novel risks across three overlapping domains—military [10], international relations [11, 2, 38], and institutional (e.g., universities, tech companies) [12, 15, 39, 40]—that differ from those associated with conventional LAWS. First, poorly validated or overtrusted systems can cause battlefield failures—including misidentification, friendly fire, brittle AI-human coordination, and performance gaps between exercises and combat [30, 41–44]. Second, unpredictable or escalatory behavior can increase the risk of interstate conflict, especially when AI systems are deployed without adequate testing or validation [45, 13, 2, 30]. Third, the rapid pipeline from AI research to military use poses risks to scientific openness, research trajectories, and the stability of international norms [46, 10, 12, 17]. We summarize these risks in Table 1.

In the **military domain**, the need to validate weapons performance during peacetime is not new—but the characteristics of modern AI make it more difficult than for conventional systems. Without additional validation, AI-LAWS are more likely to underperform in conflict, often in unintuitive ways. For example, "brittleness" is a well-documented issue in both academic and industrial AI, especially as systems become more advanced and multifunctional. It refers to the often rapid decline in performance when AI is exposed to contexts outside its training data—including variations a human would expect to be irrelevant [47, 48]. For example, an AI-LAWS trained on diverse biomes but validated only in temperate European environments may underperform in tropical or East Asian settings. This risk heightens when iterative tuning focuses on a narrow deployment context, causing

Table 2: *Examples of real-world AI-enabled military systems with publicly documented unmanned or frontier AI-centered capabilities.* Includes platforms used for targeting, coordination, or decision support across air, land, sea, and command domains. Status reflects 2019–2025 public data: In Dev = In development, Demo = Demonstrated, Deployed = In deployment.

| Domain | Name | Developer | Nation | Use Case | Status |
|---|---|---|---|---|---|
| Command / Control | Defense Llama | Scale AI | U.S. | Command, targeting, report synthesis (all systems below) | Demo |
| | Lattice | Anduril | U.S. | - | Deployed |
| | AI Platform | Palantir | U.S. | - | Demo |
| | Project Maven | NGA/DoD | U.S. | - | Deployed |
| | Hivemind | Shield AI | U.S. | Also drone swarm coordination | Deployed |
| | 18th Airborne Corps System [36] | U.S. Army | U.S. | - | Deployed |
| | Lavender [4] | IDF | Israel | - | Deployed |
| | ChatBIT [37] | PLA | China | - | Demo |
| Air | XQ-58A Valkyrie | Kratos | U.S. | Stealthy autonomous wingman | In Dev |
| | F/A-XX | Boeing, Northrop Grumman | U.S. | Optionally-manned fighter | In Dev |
| | Jetank | AVIC | China | Swarm drone carrier | Demo |
| | Saker Scout | Saker | Ukraine | Quadcopter drone | Deployed |
| | Geran-2 | Alabuga | Russia | Loitering munition | Deployed |
| | Lancet | ZALA Aero | Russia | Loitering munition | Deployed |
| | Ababil-5 | HESA | Iran | Armed ISR drone | Deployed |
| | Mohager-6 | Qods Aviation | Iran | Armed ISR drone | Deployed |
| | Kargu | STM | Turkey | Loitering munition | Deployed |
| Land | Ripsaw | Textron | U.S. | Robotic combat vehicle | In Dev |
| | LOCUST | BlueHalo | U.S. | Counter-drone laser system | Demo |
| | Uran-9 | Kalashnikov Group | Russia | Artillery UGV | Deployed |
| | Abzats | NVP Geran | Russia | Anti-drone warfare vehicle | Deployed |
| | THeMIS | Milrem | Estonia | Artillery UGV | Deployed |
| | TAIWS | Indian Army | India | Border defense robot | Demo |
| Sea | Orca XLUUV | Boeing | U.S. | Long-range submarine drone | Demo |
| | Ghost Fleet | DARPA/Leidos | U.S. | Surface vessel fleet | Deployed |
| | TRITON | Ocean Aero | U.S. | Hybrid undersea-surface, ISR and anti-submarine drone | Demo |
| | JARI-USV-A | CSSC | China | Patrol ship | Demo |

unintentional overfitting to the validation setting. As a result, performance suffers in environments the system was trained on but never field-tested in. Such failures can also be difficult to detect— the AI models AI-LAWS rely on are partially opaque, and their bases of decisions may not be interpretable or auditable with high confidence [8, 3, 14].

For the second domain, beyond battlefield performance, the development and spread of AI-LAWS may **destabilize international relations** in subtle and unpredictable ways. For example, the unpredictability of modern AI systems can create novel conflict risks if an AI-LAWS is deployed in a diplomatically tense region [13, 2]. An AI-LAWS patrolling a border may protect soldiers, but could misinterpret warning shots as hostile and initiate or recommend return fire, exacerbating tensions that could escalate to conflict. Even without deployment, uncertainty over AI-LAWS battlefield impact has fueled arms-race dynamics, prompting major investments by multiple nations [38, 21]. These investments risk becoming a security dilemma that encourages a competitive arms race. The problem is compounded by technical uncertainty around AI-LAWS capabilities and timelines, and the need to test systems in real world combat [42, 49, 41]. Lacking better insight, policymakers may overestimate their impact to avoid under-investing, while also seeking conflict to test them in, as is the case in Ukraine. These dynamics raise the risk of escalation even before AI-LAWS are deployed.

Lastly, AI-LAWS pose risks to **academic and civilian AI** freedoms and norms. Modern AI is distinctive in that the gap between civilian and military applications is unusually small. As a result, civilian advances are easily repurposed for military use, which has national and global security implications. This incentivizes governments to overregulate the process and dissemination of AI research and models, e.g., through publication blackouts and travel restrictions on AI researchers [12]. As civil-military barriers erode, regulatory oversight, export restrictions, and international controls may further limit civilian AI development. Some restrictions—such as controls on AI hardware exports—are already in place, and similar constraints on software and algorithms are likely to follow [50–52]. These risks are amplified by uncertainty. When policymakers lack clarity about AI's national security implications, they may over-regulate to hedge against worst-case scenarios. These governance failures are addressed in the next section.

*All three risk types can be mitigated—but doing so will require technical insight from AI researchers and tailored standards for validation, accountability, and research integrity.*

## 2.3 Existing policy efforts do not address unique AI-LAWS risks

**Despite broad policy attention, current governance frameworks do not account for the technical risks posed by modern AI systems**. Nations continue to field military AI, yet no binding standards exist to govern how the systems are designed, validated, or deployed [45]. Most existing LAWS and AI safety frameworks emphasize broad principles—such as human oversight or bias minimization—but lack concrete mechanisms and metrics tailored to how these systems function in practice.

Alongside these awareness issues, the AI research community has been largely isolated from military governance conversations. Initial regulatory efforts were led by defense officials, with limited technical input from the scientific community [53]. Today, oversight of military AI is debated at institutional, national, and international levels [54, 26]—yet those best equipped to evaluate system behavior remain underrepresented. Militaries worldwide have expressed concern over lacking AI expertise to update doctrine and regulation as technology evolves [15, 55–57]. This disconnect creates both risks and opportunities. To meaningfully address AI-specific failure modes, researchers must be at the table [58]. AI researchers who seek to shape policy must engage with military actors beyond their usual academic spheres [17, 15, 59]. Engagement is essential to ensure that governance reflects real-world system behavior—not just design intentions or policy abstractions.

Regulatory frameworks have not kept pace with failure modes such as opacity, drift, and escalatory misuse [21]. Technical and policy communities alike struggle to establish thresholds for oversight, especially as systems grow more adaptive, autonomous, and complex. Current certification procedures are rooted in conventional systems and offer little clarity on AI-specific metrics or standards [45, 60, 56]. These gaps are compounded by geopolitical concerns: policymakers may resist unilateral regulation for fear of losing strategic advantage in advancing AI technologies [38], and international progress has been limited by misaligned priorities and inconsistent institutional mandates [61].

Efforts to define AI-LAWS—in the United Nations [38], Track II forums [62, 63], or REAIM [64]—have struggled to yield useful standards [65]. Technical experts remain underrepresented in these discussions, and proposed definitions often hinge on extreme thresholds like "full autonomy" or "superhuman learning" [53]. One influential definition, for example, requires that a system expand its capabilities beyond human expectations without any human intervention [38]. While well-intentioned, such thresholds exclude most deployed and near-term systems—and thereby prevent oversight of the systems already in use.

# 3 Oversight Criteria for AI-LAWS, Grounded in System Behavior

Effective oversight of AI-LAWS requires moving beyond labels like "autonomous" or "general intelligence" and toward criteria grounded in how systems actually function. Targeting models can misidentify civilian vehicles in unfamiliar terrain [27, 13]. Autonomous patrols may escalate based on ambiguous signals, without human authorization [2, 13]. Command-advising models may drift over time in ways operators cannot detect [3, 44]. These risks are not hypothetical—they mirror failures already observed or anticipated in real-world systems (see Table 1). Oversight must begin with how systems behave, not how they are labeled.

Current frameworks often hinge on vague descriptors—like "human-in-the-loop" or "fully autonomous"—that fail to reflect how systems behave in practice. For example, a loitering munition may shift between human-authorized and AI-executed modes based on mission context [23]. Command-advising systems may shape lethal decisions without directly executing them [9, 3]. Despite posing real governance challenges, such systems risk not qualifying as AI-LAWS under strict definitional thresholds—even when they should.

To address this gap, **we propose a rubric based on system behavior.** Military AI systems will trigger enhanced oversight when they meet both of the following two criteria:

- **Criterion 1:** Use of AI/ML methods (e.g., neural networks) that are necessary for system function, and that pose AI-specific risks such as misidentifying targets, escalating unpredictably, drifting after deployment, or degrading in unfamiliar environments due to limited generalization beyond the training data distribution.
- **Criterion 2:** Involvement in semi- or fully autonomous targeting and force application. At least one capability that requires AI/ML contributes to decision making or advising on the use of lethal force, with some degree of autonomy and with or without human oversight.

These behavior-based criteria are designed to be tractable for policymakers and aligned with how AI-LAWS operate. They offer a practical tool for identifying high-risk systems and guiding regulatory scrutiny and validation. Crucially, our rubric creates a point of entry for AI researchers to collaborate meaningfully with military and policy stakeholders on oversight and failure mode analysis. Figure 1 summarizes our rubric and contrasts it with existing oversight narratives.

**AI-LAWS that meet these criteria exhibit unique risks.** Systems that meet both criteria are likely to introduce governance-relevant failure modes not found in traditional automation. These include undetected errors, misaligned escalation, loss of human accountability, and destabilizing institutional incentives, as described in subsection 2.2 and summarized in Table 1.

One of the main barriers to regulating conventional LAWS has been disagreement over definitions [66, 65]. According to U.S. submissions to the UN, LAWS—sometimes called "killer robots"—are defined as systems that "can identify, select, and engage targets with lethal force without further intervention by an operator" [67]. Critics note that this definition is overly broad, potentially encompassing naval mines and heat-seeking missiles. The radar-seeking Israeli Harpy loitering munition, deployed in the late 1980s, exemplifies early LAWS already covered by prior regulations as historical models behave predictably with simple tracking functions [66]. Recent advances in AI research have significantly increased the combat power and autonomy of AI-enhanced systems [68, 69, 11, 3, 5, 15]. As AI capabilities were added to LAWS, it became increasingly necessary to distinguish conventional automated weapons from AI-LAWS [23, 26, 60]. While AI has long been used in simple forms for military tasks [70, 34], **modern AI-LAWS employ far more advanced ML models, can operate along a continuum of modes from remote-controlled to fully autonomous, and present unique AI-specific risks and failure modes not seen in conventional LAWS.**

With a definition of AI-LAWS in hand, the next question is what principles should guide their governance. Figure 1 shows how our criteria relate to existing oversight narratives.

**Oversight of military AI systems has traditionally centered around three narratives.** While each raises valid concerns, none is technically grounded enough to address how deployed systems behave—particularly regarding opacity, drift, and brittleness when the domain of their use shifts. The first, rooted in humanitarian advocacy, calls for bans or strict human-in-the-loop constraints based on the principle of *Meaningful Human Control* (MHC) [71, 72, 22]. The second emphasizes long-term existential risks from future artificial general intelligence (AGI) [73, 74]. The third—most

common in defense policy—treats AI-LAWS as a natural extension of conventional automation and maintains that existing frameworks, such as the U.S. DoD's Directive 3000.09, are sufficient [60]. This approach, termed *Appropriate Human Judgment (AHJ)*, is embedded in U.S. military doctrine [60, 75]. It emphasizes procedural safeguards—such as system testing, operator training, and robust design—rather than requiring direct human control over every lethal force application. AHJ is often viewed as more pragmatic for increasingly complex systems—such as when operations require faster response times than humans can provide, involve cooperative autonomous swarms, or scale beyond what human oversight can track reasonably in real time [7].

Yet all three narratives remain underdefined and difficult to operationalize. MHC lacks measurable criteria for what constitutes "meaningful" control in fast-moving or distributed contexts. AHJ, though more scalable, often fails to define metrics for what constitutes sufficient "judgment"—or to codify how to assess the safety of opaque, adaptive, or continually learning systems [60]. Critics note that AHJ can obscure accountability behind vague assurances of system design—especially when those assurances come from third-party contractors or defense tech firms [13, 2, 17, 44, 76].

Policy limitations have real consequences. Post-deployment drift, overreliance on black-box systems, brittle generalization, and institutional pressure to avoid challenging flawed AI outputs can all occur under either oversight model [21]. Command-advising systems may formally keep humans "in the loop," yet users often defer to flawed recommendations—especially under time pressure or when authority to intervene is unclear [77, 30]. **To be effective, any oversight framework—whether based on MHC, AHJ, or another model—must be paired with rigorous behavioral validation, grounded in AI research expertise.** This includes task-specific performance benchmarks, failure audits, stress testing under domain shift, and post-deployment monitoring. These mechanisms do not replace high-level policy principles; they make them testable and enforceable. The U.K.'s 2024 guidance for military AI use offers a promising start, with proposals for AI-specific audits, operator certification, and red-line prohibitions [56]. However, as AI capabilities advance, fixed rules alone cannot keep pace. Governance must evolve alongside systems with flexible frameworks—and be rooted in how those systems behave, not just how they are categorized or intended to function.

**The United States and the United Kingdom have introduced policies to govern military autonomy, but these efforts remain incomplete.** While the U.S. Department of Defense's Directive 3000.09 and the U.K.'s 2024 guidance outline principles for AI oversight, they do not specify how those principles apply to modern, adaptive AI-LAWS. Directive 3000.09 formally applies to all LAWS, including those that incorporate AI. It articulates high-level goals—such as ensuring AHJ, resilience to adversarial attacks, and review after model updates [60]. However, it offers little guidance on what these goals mean in practice—for example, it does not define what qualifies as a "substantial"

**A. Current Landscape of Military AI Safety Policy**

"**AI-LAWS** *are a distinct new category needing oversight with AI researchers in the loop*" – Position in this paper

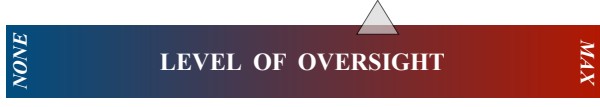

"Existing policy around autonomous weapons covers AI-LAWS sufficiently" –status quo proponents

"Blanket ban on autonomous weapons, AI or not" –humanitarian disarmament community (eg ICRC, CSKR)

**B. Criteria for AI-LAWS Oversight**

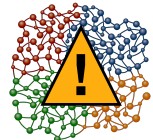 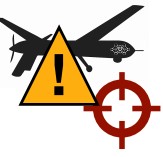

1. AI/ML methods (e.g., neural networks) are necessary for function

2. AI/ML involved in autonomous targeting & force application, with or without human oversight

Figure 1: *Military AI policy is shaped by competing narratives, but none fully address the risks posed by AI-LAWS.* **A.** A spectrum of views on autonomy ranges from blanket bans from e.g., International Committee of the Red Cross (ICRC) and Campaign to Stop Killer Robots (CSKR), to status quo arguments that existing policy around autonomous weapons systems is sufficient. We propose AI-LAWS as a distinct category requiring technically informed oversight. **B.** Our rubric identifies AI-LAWS using two criteria: (1) use of AI/ML essential to system function, and (2) involvement in semi- or fully autonomous targeting or force application, with or without human oversight. These criteria are grounded in system behavior—not labels—and define the subset of military AI systems that require new governance mechanisms with AI researchers in the loop.

model update requiring revalidation, or how often retrained or continually learning models should be re-evaluated. As a result, critics—including voices within the defense industry—warn that some AI-LAWS may enter deployment without rigorous review [45].

The U.K. framework sets a stronger foundation. Its 2024 guidance proposes certification processes, third-party audits, and red-line prohibitions on delegating specific decisions to autonomous agents [56]. However, the proposal is not a binding regulatory framework, and key questions—such as how to validate adaptive models or define escalation-related boundaries—remain unresolved. The next section helps address this oversight gap, by offering specific initial policy proposals to codify.

## 4   Policy Recommendations for AI-LAWS Oversight

The following recommendations address urgent, technically grounded failure modes already visible in real-world military AI systems. While implementation rests with national and institutional actors, each recommendation highlights an issue where AI researchers can help define performance thresholds, anticipate failure modes, and ensure oversight reflects how these systems actually function.

**1. Ban AI control over nuclear weapons deployment.**   Of all AI-LAWS use cases, nuclear command and control is the clearest red line. Some argue that delegating or augmenting nuclear command with AI could improve deterrence over ambiguous or manually executed threats [78, 79]. This view is extremely dangerous. Systems that delegate launch or targeting decisions in nuclear scenarios pose disproportionate risks—due to the stakes, the execution speed, and the unpredictability, opacity, and drift of AI under real-world conditions. Recent advances in AI speed and complexity have amplified the risk of escalation—via misclassification, drift, or adversarial manipulation [2, 14, 80]. Yet tests of AI-based nuclear advisors date back decades [34].

We support formal, binding prohibitions on AI system control over nuclear deployment pipelines. Existing proposals—including bilateral U.S.–China talks and a 2023 U.S. Senate bill [81, 82]—offer strong starting points, but lack international commitment and technical specificity. The 2022 National Defense Strategy committed the U.S. to require a human in the loop for any nuclear launch [83], but made no mention of AI advising systems. In 2023, the P3 (U.S., U.K., and France) issued more AI-specific declarations in the P5 council, urging China and Russia to follow suit [84].

*Key message: Future policy should explicitly ban AI-based military systems from making nuclear launch decisions, regardless of human oversight. AI advising on nuclear strategy should either be banned outright or tightly constrained with technical specificity* [2, 14, 44].

**2. Develop international standards for AI-LAWS validation.**   Many risks from AI military systems stem from insufficient validation—such as poor generalization performance, adversarial vulnerability, and low human interpretability. These challenges are well known in civilian AI [43], but their consequences are amplified in combat settings. Without shared standards for evaluating AI-LAWS under real-world constraints, oversight remains inconsistent and easily circumvented.

We recommend developing international validation protocols tailored to AI-LAWS. Unlike past proposals that would have mandated fixed oversight models (e.g., hard human-in-the-loop controls for low-level decisions [71, 85]), we do not advocate a one-size-fits-all standard. Instead, we call for a coordinated international process to define what "sufficient oversight" means—anchored in technical performance rather than philosophical abstraction. This effort should include behavioral benchmarks, context-shift performance thresholds, and documentation of failure modes that may emerge only after deployment. Real-world performance should take precedence over interpretability, which remains poorly standardized for complex models [86–89]. The goal is not to constrain architectures, but to ensure AI-LAWS reliably identify targets, adapt safely, and behave predictably under stress. Notably, setting such standards can be done without disclosing sensitive details on AI-LAWS, and individual governments would handle the actual validation process using their own personnel. Several countries and organizations have proposed promising components of such a framework [56, 60, 90, 54], which should now be formalized, stress-tested, and updated in collaboration with AI researchers. AI researchers can help shape criteria, vet claims, and clarify which risks remain invisible by default.

While it is not pragmatic nor desirable to encourage hostile nations to share knowledge for making more lethal and dangerous weapon systems, it is worth establishing an international forum or agency to develop evolving best practices in validating military AI. Indeed, there are already considerations in the UN for establishing a *"global AGI observatory, certification system for secure and trustworthy AGI, a UN Convention on AGI, and an international AGI agency"* for parallel concerns in the domain of hypothetical AI with superhuman intelligence [91]. However, unlike AGI/ASI, dangerous and

impactful military AI already exists, yet no specialized observatory, convention or agency is being considered to handle the AI-specific concerns around frontier military AI systems, if they do not meet the difficult-to-quantify threshold for AGI/ASI [92, 93].

We propose a voluntary international consortium where participating states collaboratively define and refine AI-LAWS oversight standards. Membership in this consortium would grant representation in shaping criteria—ensuring buy-in, legitimacy, and flexibility across political and technical domains. Standards would evolve iteratively—based on new deployments, emerging failure modes, and evolving technologies—much like internet protocols, cybersecurity, or global financial risk auditing systems [94–97]. This model balances sovereignty concerns with the shared responsibility to rigorously manage the development of AI-LAWS given the potential risks. It also avoids the rigidity of treaty law while creating a durable mechanism for international norm-setting. In practice, the consortium could coordinate audits, verify compliance, and serve as a shared reference point for certifying AI safety in defense systems.

*Key message: Effective AI-LAWS oversight must be globally coordinated yet technically flexible. An international standards body is essential to ensure governance keeps pace with evolving capabilities.*

**3. Ban AI systems that direct the actions of human soldiers ("AI generals").** Recent proposals have explored using large language models and other generative AI systems to direct human combatants—a concept sometimes referred to as "minotaur warfare" [98, 99]. In this model, AI systems act as battlefield commanders—issuing orders, making strategic recommendations, or coordinating multiple units. While such approaches are technologically novel, they introduce significant risks with little demonstrated benefit as the scale of the AI command increases and human oversight decreases. Some scholars argue that AI may actually impose a requirement for greater human oversight [100].

Generative and decision-support AI systems often exhibit hallucinations, drift, adversarial vulnerability, and misplaced confidence [17, 101–105, 44, 43]. These characteristics are particularly dangerous in high-stakes, real-time decision contexts like military command [77, 106, 17, 14]. Equally concerning is growing evidence that humans tend to overtrust such systems—especially when they appear authoritative, sycophantic or fluent [30, 107–109]. In command settings, this risk can distort accountability, suppress dissent, and create institutional overreliance on opaque AI systems. This is not a call for Luddism. It is a recognition that delegating command decisions amplifies risks across all three domains: field-level failure, escalatory instability, and institutional erosion. We therefore recommend a categorical ban on AI systems that autonomously command human soldiers with minimal human oversight—replacing human officers with AI officers.

*Key message: Command decisions are among the most consequential and context-sensitive elements of warfare, and should remain under meaningful human control.*

**4. Clarify the legal status of civilian AI infrastructure under international humanitarian law.** As military systems increasingly rely on publicly available AI infrastructure—cloud compute, foundation models, and research personnel—the line between civilian and military assets is becoming dangerously blurred. Under Protocol I of the Geneva Conventions, civilian objects become lawful military targets if they make an effective contribution to military action. Yet it remains unclear how this doctrine applies to AI systems—such as models, training data, and compute infrastructure—used in both civilian and military contexts. Without clear guidance, a wide range of actors—including universities, commercial labs, and even individual researchers—could become valid military targets if their work is repurposed into AI-LAWS [110]. This creates unacceptable ambiguity for technical personnel who may unknowingly contribute to military systems. It also raises serious safety and ethical concerns for companies and institutions that host dual-use models or infrastructure. For example, one of the bombing targets in the brief 2025 Iran-Israel conflict was the Weizmann Institute of Science which houses frontier AI research labs [111]. While it's not known whether the AI labs were the primary motivation for the bombing, it's suggestive that civilian research centers may be targeted more frequently as AI becomes more central to warfare. We recommend international treaties or norms that explicitly define when and how AI infrastructure—models, data, compute platforms, and personnel—qualifies as a military objective. This includes dual-use thresholds and legal boundaries for open-source contributions.

*Key message: These definitions are critical for reducing escalation risks and protecting the integrity of civilian AI research.*

**5. Establish institutional-level civil-military boundaries in AI research and infrastructure.** The increasing convergence between civilian AI research and defense institutions—via funding,

infrastructure, and personnel—is eroding long-standing boundaries between the two spheres [17, 15, 112, 40, 39]. This shift often occurs through formal programs such as dual-use labs or military contracting. Examples include proposals from the U.S. Defense Innovation Unit (DIU) to embed military AI research centers in universities [15], and China's civil-military fusion strategy, which integrates academic and defense institutions [33, 113]. In both cases, research departments may become partially governed by national security interests, affecting what can be published, who can be hired, and how international partnerships function. While some collaboration is inevitable, blurred boundaries in historically civilian institutions risk undermining academic freedom, reducing international cooperation, and deterring researchers from working on civilian or humanitarian goals [31, 32, 12]. These shifts create a research ecosystem where technical expertise in AI is increasingly channeled toward military applications—sometimes without clear ethical guardrails or institutional independence. It may also introduce new AI safety risks, as rapid militarization of foundation models incentivizes premature deployment—especially in closed or classified settings [3, 44].

We recommend that institutions and companies publicly declare their own guidelines for preserving the independence of civilian AI research, since no one-size-fits-all policy can apply across sectors or nations. These declarations were once common, but many institutions have quietly moved away from them [114–116], creating confusion and concern among employees [117, 118]. Declarations should include disclosure of military funding and affiliations, and clarifications of any commitments to separating civilian and military personnel, models, infrastructure, and funding—aligned with institutional goals and protections. Academic institutions should safeguard research agendas from redirection during reorganizations tied to defense contracting. They should also protect students and early-career researchers from being drawn into classified work without clear opt-in pathways. Review boards could be considered to assess the civil-military implications of AI projects—similar to existing models for biosafety and human subjects research. These structures can help ensure that dual-use concerns are addressed without restricting scientific inquiry or leading to unacceptable risks.

*Key message: Civilian AI research should be protected by clear institutional boundaries and opt-in safeguards—not shaped by defense interests through ambiguity.*

## 5  AI researchers can contribute on military AI policy

It is important to note that there are a number of means for academic and civilian researchers to contribute to military AI regulation, and that a career in policy or national security work is not required to be involved. Ways for AI researchers to get involved in these discussions include, but are not limited to: (1) Participate in international conferences or dialogues on AI safety and arms control [64, 119–121]. (2) Publish research on AI red teaming and failure modes in contexts relevant to the military [2]. (3) Collaborate with researchers at think tanks, policy institutes or military colleges which do public-facing policy research relevant to military AI [122, 123]. (4) Encourage universities and departments to establish written norms and policies defining boundaries for military AI research, as institutions in the U.S. and Europe generally lack explicit guidelines on acceptable defense-related topics [124–126]. (5) Avoid overstating benchmark results. Researchers should recognize that both think tanks and military analysts closely monitor technical AI publications. It is important that our community does not overstate benchmark success as implying immediate readiness for complex real world applications, which could encourage rushed and risky adoption by non-experts [43, 127, 128].

## 6  Conclusion

Integrating frontier AI algorithms into weapon systems has created AI-LAWS—a class of system that behaves differently from conventional LAWS and demands new governance. These systems pose structural risks—such as unpredictability, opacity, and post-deployment adaptation—that existing oversight frameworks are not designed to address. Regulatory discussions have been slowed by vague definitions, abstract debates, and—most critically—a lack of engagement from AI experts. We introduced two criteria as a tractable basis for determining when AI-LAWS warrant oversight. They provide a foundation for targeted policy and validation standards. Our recommendations offer a practical starting point for AI-LAWS regulation, which is grounded in current and near future capabilities and aligned with technical foundations and realities. Realizing these proposals will require deeper engagement from AI researchers—especially in defining performance thresholds, stress-testing deployed systems, shaping validation practices grounded in real-world behavior and establishing AI safety principles for the military domain. Military AI governance is a deeply technical challenge that demands scientific input, and AI researchers are uniquely positioned to help ensure these systems are evaluated, constrained, and held accountable.

# 7 Acknowledgments

We thank Nilufer Jason Mistry Sheasby, Ahmed Mehdi Inane, Shayne Longpre, and Flavio Martinelli for their helpful comments and feedback during the writing of this piece.

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
