# OpenReview forum: "Military AI Needs Technically-Informed Regulation to Safeguard AI Research and its Applications"
_NeurIPS.cc/2025/Position_Paper_Track — NeurIPS 2025 Position Paper Track_

### Official Review · Reviewer_D4eN · 2025-08-01

**Significance:** 3
**Presentation:** 3
**Rating:** 5
**Confidence:** 4

**Summary:**

This paper introduces the concept of AI-powered Lethal Autonomous Weapon Systems (AI-LAWS) and argues that these systems, distinguished by their reliance on machine learning for targeting, decision-making, or coordination, pose unique and urgent risks that current legal and policy frameworks fail to address. These risks include brittle generalization, opacity in decision processes, escalation in conflict zones, and the erosion of scientific openness due to civil-military entanglement. The authors propose a behavior-based definition for AI-LAWS, grounded in their actual function rather than vague notions like “full autonomy” and advocate for technically informed governance. They offer concrete policy recommendations such as banning AI control over nuclear weapons, establishing international validation standards, rejecting AI-based battlefield command (“AI generals”), and safeguarding civilian AI infrastructure and academic independence. The main argument of the paper is that current policies for regulating autonomous weapons fail to address the real-world risks introduced by modern AI systems, and that regulation must be grounded in technical behavior with direct involvement from AI researchers.

**Strengths:**

This paper excels in presenting a timely, well-structured, and technically grounded argument for the need to regulate AI-powered lethal autonomous weapon systems (AI-LAWS). It clearly defines its terms, introduces a behavior-based oversight framework, and supports its claims with concrete examples of deployed systems, real-world risks (e.g., drift, brittleness, misclassification), and detailed citations from both the AI and policy literature. The paper is especially strong in connecting AI technical failure modes to geopolitical and institutional consequences, making a compelling case for the inclusion of AI researchers in the regulatory lifecycle. The topic is highly relevant to the NeurIPS community, as it addresses the dual-use nature of ML research, the erosion of academic openness, and the ethical obligations of researchers. Its recommendations are actionable, its structure is clear, and it meaningfully engages with alternative views. Overall, it makes an impactful and well-supported contribution to a critical and underdiscussed area.

**Weaknesses:**

While the paper presents a strong position, several areas could be clearer. The proposed oversight criteria for AI-LAWS, based on whether a system uses ML and contributes to lethal decisions, are useful but not supported by detailed examples of how to detect or verify these conditions in real systems. The rubric would be stronger if applied to specific systems in Table 2 to show how it distinguishes high-risk cases. The paper also states that AI researchers should be involved in oversight but does not explain how this would happen, e.g., through system testing, access to classified models, or formal review roles. In several places, the risks discussed (e.g., opacity, drift, brittleness) are technically valid, but the paper could clarify how they differ from risks in non-lethal AI systems.

**Questions:**

The paper raises important points, but several areas would benefit from further clarification and development to strengthen its practical impact and address key ethical concerns.
-- Recommendation 1: Clarify how the behavioral criteria for AI-LAWS can be applied in real systems and how evaluators should assess risks like misclassification, drift, or escalation.
-- Recommendation 2: Provide a concrete example from Table 2 to show how the rubric identifies a system as an AI-LAWS.
-- Recommendation 3: Address how the framework applies to systems that influence but do not directly execute lethal actions.
-- Recommendation 4: Specify the role AI researchers are expected to play in oversight, including the type of access or responsibilities required.
-- Recommendation 5: Explain why risks such as bias, opacity, and brittle generalization demand separate treatment in military AI systems compared to civilian ones.

**Alternative Position:**

Yes, and alternative positions are well-considered and addressed by the argument

**Author Identification:**

No.

**Context:**

3

**Details Of Ethics Concerns:**

This paper raises serious ethics concerns related to the use of AI in military systems. It discusses AI-powered lethal autonomous weapon systems (AI-LAWS) that can make or influence life-or-death decisions in real time, often in unpredictable or adversarial environments. These systems present substantial safety and security risks, including the potential for misclassification, drift, brittleness under domain shift, and escalation without human intervention. There are also clear human rights implications, as AI-LAWS may be deployed in surveillance or targeting roles that violate international humanitarian law and put civilians or civilian infrastructure at risk. Furthermore, the paper highlights issues of bias and fairness, noting that poorly trained or narrowly validated models may misidentify targets, especially in underrepresented geographies or populations. Finally, it underscores data quality and generalization concerns, where failures stem from models trained in narrow contexts that perform unreliably in the field. These ethical issues are not theoretical but emerge from documented deployments and real-world use cases, making them central to the paper’s argument for technically informed regulation.

**Discussion:**

1

**Ethics:**

["Major Concern: Data quality and representativeness", "Major Concern: Safety and security", "Major Concern: Discrimination, bias, and fairness", "Major Concern: Human rights (including surveillance)"]

**Position:**

Yes, the paper argues for or against a position related to machine learning.

**Support:**

3

**Thoroughness:**

5

---

### Official Review · Reviewer_4P3C · 2025-08-12

**Significance:** 4
**Presentation:** 3
**Rating:** 9
**Confidence:** 5

**Summary:**

The paper defines AI-powered lethal autonomous weapon systems (AI-LAWS) as military systems using modern AI/ML that present unique risks beyond conventional LAWS. It proposes criteria that makes a military AI systems high-risk such as essential AI/ML components and autonomous targeting/force application. It calls for technically informed regulation with AI researchers involved. The paper also recommends targeted policies: banning AI in nuclear launch and battlefield command, setting international validation standards, clarifying civilian AI infrastructure’s legal status, and preserving civil-military boundaries.
The paper has a clear position: AI-LAWS are a distinct new category needing oversight with AI researchers in the loop.

**Strengths:**

- The argument is well-structured, moving from problem framing to a practical oversight rubric and targeted policy recommendations.
- The paper clearly identified the risk of AI lethal weapons systems, and showed examples of real-world use in Tables 1 and 2.
- The paper effectively distinguishes AI-LAWS from conventional autonomous weapons.
- The topic is highly relevant to the NeurIPS community and critical given the current state of geopolitics around the world.

**Weaknesses:**

- The proposed definition and criteria of AI-LAWS is clear as a concept, but lacks concrete metrics, or examples of how it would be enforced in real-world policy.
- The paper presented the current existing frameworks as alternative views, but it should have also discussed opposing views that discuss infeasibility of regulating AI-LAWS or views of defense stakeholders who may see current oversight as sufficient.
- The paper doesn't clearly specify the role of AI researchers for each policy recommendation they present.

**Questions:**

- How do you plan to translate the definition into measurable benchmarks that can be applied consistently across different geopolitical and security contexts to classify AI-LAWS?

- Among the listed risks which should be addressed first in regulation and how should limited oversight resources be allocated?

 - How effective is the proposed approach compared to just integrating AI-LAWS oversight into broader military AI governance frameworks, rather than treating them as a distinct category?

**Alternative Position:**

Yes, and alternative positions are well-considered and addressed by the argument

**Author Identification:**

No.

**Context:**

4

**Discussion:**

4

**Ethics:**

["NO or VERY MINOR ethics concerns only"]

**Position:**

Yes, the paper argues for or against a position related to machine learning.

**Support:**

4

**Thoroughness:**

4

---

### Official Review · Reviewer_YQGe · 2025-08-14

**Significance:** 4
**Presentation:** 4
**Rating:** 7
**Confidence:** 4

**Summary:**

The paper’s focus is an emerging technology referred to as AI-LAWS: lethal autonomous weapons that integrate AI. The authors argue that current governance efforts are insufficient for AI-LAWS: they require a distinct regulatory regime from conventional LAWS, into which AI experts are substantially integrated. This need stems from novel risks – including opacity, post-deployment drift, brittleness under distribution shift, miscalculation/escalation, and erosion of human oversight – that are both poorly addressed by current frameworks and poorly understood by those lacking AI expertise. To resolve definitional ambiguity, the authors propose a shift to behavior-based criteria for identifying AI-LAWS, and advocate for a slew of policies for the oversight of the systems that meet those criteria: (1) ban AI control of nuclear deployment; (2) develop more rigorous international standards for technical validation AI-LAWS via a voluntary international consortium; (3) ban “AI generals”; (4) clarify the legal status of dual-use AI infrastructure; and (5) establish greater institutional separation between civil and military AI.

**Strengths:**

The authors successfully establish the importance and timeliness of addressing AI-LAWS oversight, and sharply articulate associated risks, providing a compelling case for why a new governance regime is needed vs. conventional LAWS. Technical aspects like opacity and brittleness under distribution shift are clearly explained and make a strong case for the integration of technical expertise, alongside an explicit and concrete validation agenda (out-of-distribution robustness, red-teaming, end-to-end trials). Proposed prohibitions (against AI in nuclear deployment and “AI generals”) seem sufficiently narrowly drawn and potentially politically feasible. The topic is highly relevant to the NeurIPS community, especially in light of the papers’ call for the participation of ML experts in setting validation criteria and standards for AI-LAWS.

**Weaknesses:**

The proposed behavior-based definition of AI-LAWS does seem like a step toward greater clarity, but the authors could have provided greater justification for why this definition is sufficiently operational. In particular, the description of Criterion 1 as simply the use of those AI methods “that pose AI-specific risks” may be underspecified, conflicting with the authors’ claim that their definition offers a “practical tool for identifying high-risk systems.”
The authors’ proposal of international standards creation via voluntary consortium as opposed to treaties seems novel and interesting for discussion, but likely raises questions about incentives, enforceability, and auditability that remain under-addressed in the paper.
While recommendations are actionable, perhaps the paper’s most central thesis – that greater AI expertise ought to be integrated into AI-LAWS oversight – remains at times vague. Greater attention could be paid to institutional/governance structures, and specific recommendations could be provided for ensuring that technical expertise is involved at all levels.

**Questions:**

What mechanisms do you foresee for the enforcement of standards set by a voluntary consortium? How often do you expect revisions to standards to happen? How to integrate this into existing fora?

**Alternative Position:**

Yes, and alternative positions are well-considered and addressed by the argument

**Author Identification:**

No.

**Context:**

3

**Discussion:**

3

**Ethics:**

["NO or VERY MINOR ethics concerns only"]

**Position:**

Yes, the paper argues for or against a position related to machine learning.

**Support:**

3

**Thoroughness:**

4

---

### Note · Authors · 2025-09-03

**1-10 Additional Comments:**

The character limit for these responses and for the reviews should be increased somewhat- it can be hard to discuss technical details with a very small response limit.

**1-11 Submit Again:**

Definitely yes

**1-1 Submission Process:**

4

**1-2 Next Year:**

We think that a more standard position paper rebuttal such as how ICML handles their position track works better. It is confusing to have a reduced version of the rebuttal merged with a reviewer feedback assessment.

**1-3 Future Development:**

Closer coordination with the main track on deadlines and review load- Most of the period for doing reviews overlapped with the rebuttal/discussion period for the main track, which was hectic to juggle and meant reviewing an abnormally large number of papers for one conference.

**1-4 Interest:**

["Panel discussions with other position paper authors", "Structured debates on controversial topics", "Workshops for developing position papers", "Mentorship programs for early-career researchers", "Other (please specify in the next question)"]

**1-4 Other Interest:**

Some form of workshops/discussions/talks/etc aimed at bringing NeurIPS position paper ideas and arguments to a non-technical audience within the space of the broader conference, especially to address policy-related audiences.

**1-5 Thoughtful:**

8

**1-6 Supportive:**

7

**1-7 Technical Aspects Versus Position:**

7

**1-8 Gate Keeping:**

10

**1-9 Camera Ready Changes:**

(Change 1: Deepen discussion about ways for AI experts to get involved in military AI regulatory discussions)

We will clarify some specific examples to help readers. These examples will span: (1) current related Track 2 dialogues and international conferences such as REAIM, (2) non-profits that fund work into this topic such as Open Philanthropy, and (3) International organizations that are relevant to the topic (proposed AGI UN agency, ICRC, Convention on Certain Conventional Weapons, etc).

(Change 2: Discuss more table 2 examples in the context of our new definition and risk criteria)

We will expand the section, and discuss more specific models in different military domains, as well as the degree to which it is known that AI and autonomy is incorporated into their functions.

(Change 3: Discuss whether the AI/ML community should assist in making any benchmarks designed specifically for military AI safety purposes)

We will emphasize the importance for ML/AI researchers to not overstate current benchmarks in any computer science domain with potential military use, which would risk leading military officials over-optimistic recklessness about transferring current AI models to military applications.

We also believe it would be useful for civilian scientists to have dialogue with international organizations or arms control experts to, for example, think of red teaming exercises or benchmarks that could show how civilian failure modes could predict military ones.

(Change 4: More clearly emphasize that there are productive steps for scientists to make in governance)

We will point to examples such as international AI safety conferences and Track 2 dialogues including the ongoing Brookings-Tsinghua Track 2 dialogues and the recent AI Safety Summit in Seoul. For the national government, in the US for example, AAAS and other fellowships also exist for faculty and postdocs to directly work with policy makers, and similar programs exist in Europe.

**3-1 Review Response1:**

YQGe

**3-2 Reaction To Review1:**

The reviewer gave high quality feedback and read the paper carefully in our estimation. Their comments were useful for both honing the position argument as well as strengthening technical evidence, in a balanced way. We did not feel that they attempted to gate keep us in any substantial way.

**3-3 Review Response2:**

4P3C

**3-4 Reaction To Review2:**

This reviewer seemed to approach our work thoughtfully, and properly understood our position and arguments. They seem strongly supportive of our position, but also provided several relevant and useful critiques that we will use to improve the paper (as per 1-9 above). The reviewer seems to have considered both position and technical support in their review. We do not feel they have attempted to gatekeep us.

**3-5 Review Response3:**

D4eN

**3-6 Reaction To Review3:**

This reviewer likely made a mistake in the score they assigned to our discussion potential, as described in 2-3. Other than that, the review is thorough and thoughtful, and seems supportive of our position in principle. The review focused on technical aspects of the paper predominantly. We did not feel gate-kept by this review.

---

### Meta-Review · Area_Chair_gwPM · 2025-09-13

**Rating:** 7
**Confidence:** 4

**Strengths:**

YQGe emphasises importance, timeliness, relevance, precise and compelling articulation of the need for a new framework for AI-LAWS vs conventional LAWS. Technical aspects are well explained, proposals are tractable.

4P3C likewise praises the paper's structure, clarity, and demonstration of real-world use (I also found this aspect of the paper compelling). Also praises relevance for NeurIPS community.

D4eN's praise for the paper is similar, adding positive observations about the integration of technical and regulatory themes.

**Weaknesses:**

YQGe questions the degree to which the definition of AI-LAWS succeeds, given its reliance on quite a vague and (I would add) ad hoc list of specific behavioural concerns. I also share this reviewer's concern about the challenges of any attempt to achieve international coordination around weapons systems that some actors perceive to give them a strategic or tactical advantage.

4P3C observes that operationalising the definition of AI-LAWS might be easier with concrete metrics; questions the lack of presentation of alternative views, and notes that the invocation of the importance of AI researchers to AI-LAWS policy is a rather vague call to action. I agree on this (though I'd note that the discussion of the challenges posed by the porous boundaries between civilian and military AI research is innovative and noteworthy). D4eN raises essentially the same concerns as 4P3C.

**Questions:**

One possible question here is: what if an ML system in each of the respects identified in criterion 1 proved better than some human baseline, but had some additional failure mode not in this list? Relying on current failure modes to define these systems seems like selecting on the dependent variable.

**Ethics:**

D4eN Highlights ethical issues but is not asserting that the paper itself raises ethical concerns, but that it is about an ethically relevant topic, so the correct response here is that there are no ethical violations or concerns raised *about this paper*.

**Thoroughness:**

3

---

### Decision · Program_Chairs · 2025-09-26

Accept